REGISTERED REPORT PROTOCOL

# Digitalization of adverse event management in oncology to improve treatment outcome— A prospective study protocol

**Angelika M. R. Kestler[1]☯, Silke D. Kühlwein[2]☯, Johann M. Kraus[2]☯, Julian D. Schwab🅐[2], Robin Szekely[2], Patrick Thiam[2], Rolf Hühne[2], Niels Jahn[2], Axel Fürstberger[2], Nensi Ikonomi🅐[2], Julien Balig[2], Rainer Schuler[2], Peter Kuhn[3], Florian Steger[4], Thomas Seufferlein[1]‡, Hans A. Kestler🅐[2]‡ ***

**1** Department of Internal Medicine I, University Hospital Ulm, Ulm, Germany, **2** Institute of Medical Systems Biology, Ulm University, Ulm, Germany, **3** Comprehensive Cancer Center, University Hospital Ulm, Ulm, Germany, **4** Institute of the History, Philosophy and Ethics of Medicine, Ulm University, Ulm, Germany

☯ These authors contributed equally to this work.
‡ These authors also contributed equally to this work.
* hans.kestler@uni-ulm.de

This is a Registered Report and may have an associated publication; please check the article page on the journal site for any related articles.

## Abstract

The occurrence of adverse events frequently accompanies tumor treatments. Side effects should be detected and treated as soon as possible to maintain the best possible treatment outcome. Besides the standard reporting system Common Terminology Criteria for Adverse Events (CTCAE), physicians have recognized the potential of patient-reporting systems. These are based on a more subjective description of current patient reporting symptoms. Patient-reported symptoms are essential to define the impact of a given treatment on the quality of life and the patient's wellbeing. They also act against an underreporting of side effects which are paramount to define the actual value of a treatment for the individual patient. Here, we present a study protocol for a clinical trial that assesses the potential of a smartphone application for CTCAE conform symptom reporting and tracking that is adjusted to the standard clinical reporting system rather than symptom oriented descriptive trial tools. The presented study will be implemented in two parts, both lasting over six months. The first part will assess the feasibility of the application with 30 patients non-randomly divided into three equally-sized age groups (<55years, 55-75years, >75years). In the second part 36 other patients will be randomly assigned to two groups, one reporting using the smartphone and one not. This prospective second part will compare the impact of smartphone reported adverse events regarding applied therapy doses and quality of life to those of patients receiving standard care. We aim for early detection and treatment of adverse events in oncological treatment to improve patients' safety and outcomes. For this purpose, we will capture frequent adverse events of chemotherapies, immunotherapies, or other targeted therapies with our smartphone application. The presented trial is registered at the U.S. National Library of Medicine ClinicalTrials.gov (NCT04493450) on July 30, 2020.

**Data Availability Statement:** All relevant data from this study will be made available upon study completion. However, links to the Nemo smartphone app and Desktop apps are as follows: Nemo smartphone app: https://tinyurl.com/nemoappios and https://tinyurl.com/nemoappandroid Desktop-App: https://github.com/sysbio-bioinf/NEMODesktop.

**Funding:** The funding to establish the smartphone and desktop application was provided by the Ministry of Science and Art Baden-Württemberg (Zentrum für Innovative Versorgung, ZIV), the German Federal Ministry of Education and Research (BMBF) as part of the DIFUTURE project (Medical Informatics Initiative, grant number 01ZZ1804I), and the Ministry of Social Affairs and Integration Baden-Württemberg as part of the project feelBack (networked, digital, patient-related feedback psycho-oncology). However, the conduction of the here described study receives no funding.

**Competing interests:** The authors have declared that no competing interests exist.

# Introduction

Tumor treatments are frequently accompanied by the occurrence of adverse events. Thus, the acceptance of tumor therapy (compliance) is decisively determined by the occurrence and duration of side effects. Besides, quality of life will improve, if the side effects during therapy are reduced. Accurate and early detection of therapy-associated side effects is crucial for an adequate reaction by physicians. This issue is of particular interest not only for the application of chemotherapies but also for therapies with immune-checkpoint inhibitors (ICIs) or other targeted therapies, which are increasingly used in the last years [1]. Especially with ICIs combination therapies, adverse events seem to occur early during treatment and evolve rapidly [1]. It follows, that an early perception of side effects is determinant in the decisions regarding therapy dosage (maintenance or reduction) or therapy discontinuation [1]. However, physicians only see their outpatients at specified times, e.g., every 2–3 weeks. During this time, patients often forget important information about side effects, wellbeing or are afraid to report side effects in order not to compromise the outcome of their treatment. In the clinical routine, the Common Terminology Criteria for Adverse Events (CTCAE) is the standard system to report side effects during cancer treatment [2, 3]. It scores the encountered adverse events from mild (grade 1) to life-threatening (grade 4) and death (grade 5) [4]. The CTCAE scoring is performed by physicians and depends on the patient-reported symptoms. The symptom assessment of patient and physician can be discrepant [5].

Medicine has already recognized the potential of patient-reported outcomes (PROs) during treatments. First feasibility studies reported that patients are willing to participate and most (of them) completed given questionares [2, 6–9]. Paper-based or digital questionnaires do not differ in there performance [2, 10, 11], however, patients prefer digital reporting systems [8].

Following the CTCAE, the National Cancer Institute (NCI) has developed a PRO-CTCAE [12]. This scoring system requests symptoms according to attributes like severity (none, mild, moderate, severe, very severe) or frequency (never, rarely, occasionally, frequently, almost constantly) [4, 12] and has a standard recall time of seven days [2]. Indeed, the scoring between clinicians and patients differs somewhat [13, 14]. Here, the patient-reported symptoms are more correlated with quality of life while physicians' scoring associates more to clinical outcomes or emergency admission [5, 13–15]. One reason for these discrepancies may be the differences in describing the severity of adverse events. For instance, the CTCAE scoring classifies diarrhea according to the number of stools per day (CTCAE v5.0 Clean, Tracked, and Mapping Document). In contrast, the PRO-CTCAE Measurement System requests this issue in accordance with frequency. However, frequency or severity are very subjective perceptions [14] that also depend on experiences made during the treatment [7]. Hence, it is not uncommon for physicians to assess the adverse reactions reported by the patient differently form those perceived by the patient subjectively [13, 14].

Here, we present a new and easy to use digital patient reporting system for adverse events in oncology for smartphones called NEMO (German, **Ne**benwirkungs-**m**anagement **O**nkologie). It is closer to the CTCAE scoring system than the PRO-CTCAE. NEMO provides a more detailed description of scores to get a more precise view on the adverse events and can support personalized adverse event management. Besides, the app combines questionnaires on side effects of (combination-) chemotherapies, immunotherapies and targeted therapies allowing patients with different combinations of treatments to use the app. Especially by incorporating new therapeutic approaches such as immuno- and targeted therapies, the app goes beyond all previously established evaluation systems to the best of our knowledge.

Since cancer diseases often occur at an older age [16], NEMO was specially adapted to the needs of the older population. For the development of NEMO, attention was paid to a clear

design and acoustic support to make the application attractive to all age groups [17–20]. The counterpart of the smartphone application is a desktop application that provides fast visualization of occurred adverse events for physicians. With these applications, we aim to support standardized reporting and early detection of adverse events. Thereby, we hope to improve the therapeutic safety of oncological therapies and to improve the care of outpatients. All experiences with the NEMO application in the first part will be used to improve the app for the subsequent second part. Here, it will be assessed whether patients who report their adverse events via NEMO will have reduced side effects and can be treated for a more extended period than standard procedures. The trial is registered at the U.S. National Library of Medicine Clinicial-Trials.gov under the accession number (NCT04493450).

## Materials and methods

### Implementation of smartphone and desktop application

The non-commercial and freely available smartphone application for Android and iOS was created using the open-source framework (NativeScript) licensed under the Apache 2.0 software license. For the construction of the desktop application, the open-source framework Electron was used. The query browser version for the use in emergency admission was created with Node.js and Express.

### Trial design

This study will be performed in two parts. The first part should evaluate the feasibility and acceptance of a smartphone-based adverse reaction documentation. It will be followed by a second part which will compare the impact of smartphone reported adverse events regarding applied therapy doses to those of patients with standard care.

For the detection of occurred adverse events at an early stage, the participants will answer a standardized questionnaire via the NEMO app every day. Depending on the underlying therapy scheme, the questionare in NEMO can be selected for chemotherapy (S1 Table) or combined immunotherapy, targeted therapy and chemotherapy (S2 Table). NEMO allows for switching the therapeutic settings in case the therapeutic scheme changes. During therapy sessions, the documented data is transmitted securely via QR-codes to the treating physician and used for future therapy settings.

In the first part (feasibility), 30 patients will be divided non-randomly into three age groups ($< 55$ years, 55–75 years, $>75$ years) of 10 patients each by quota sampling. These people will be followed over six months. Participants will answer the respective NEMO questionaire (S1 and S2 Tables) about adverse events on a daily basis over these six months. After six months, they will be asked to answer a questionnaire about user-friendliness and feasibility (S3 Table). Evaluation of these questionnaires allows for studying the different patients' characters and their potential benefit of using NEMO. The answered questionnaires could also give insights to identify barriers to answer the questions. The results of this evaluation will enable us to amend the criteria for recruiting patients for the second part of our study, such as their age and to improve the NEMO application.

In the second part (reduction of adverse events), 36 participants will be randomly assigned to one of two groups of 18 subjects each (group 1: smartphone user and standard care, group 2: standard care), see Fig 1. Again, smartphone users will answer their respective NEMO questionnaires every day (S1 and S2 Tables) about occurred adverse events for six months. The transfer of their reported data takes place every second month in parallel with the reevaluation of tumor therapy. Participants of the standard care group will report their occurred adverse events and duration only during their personal appointments. What the

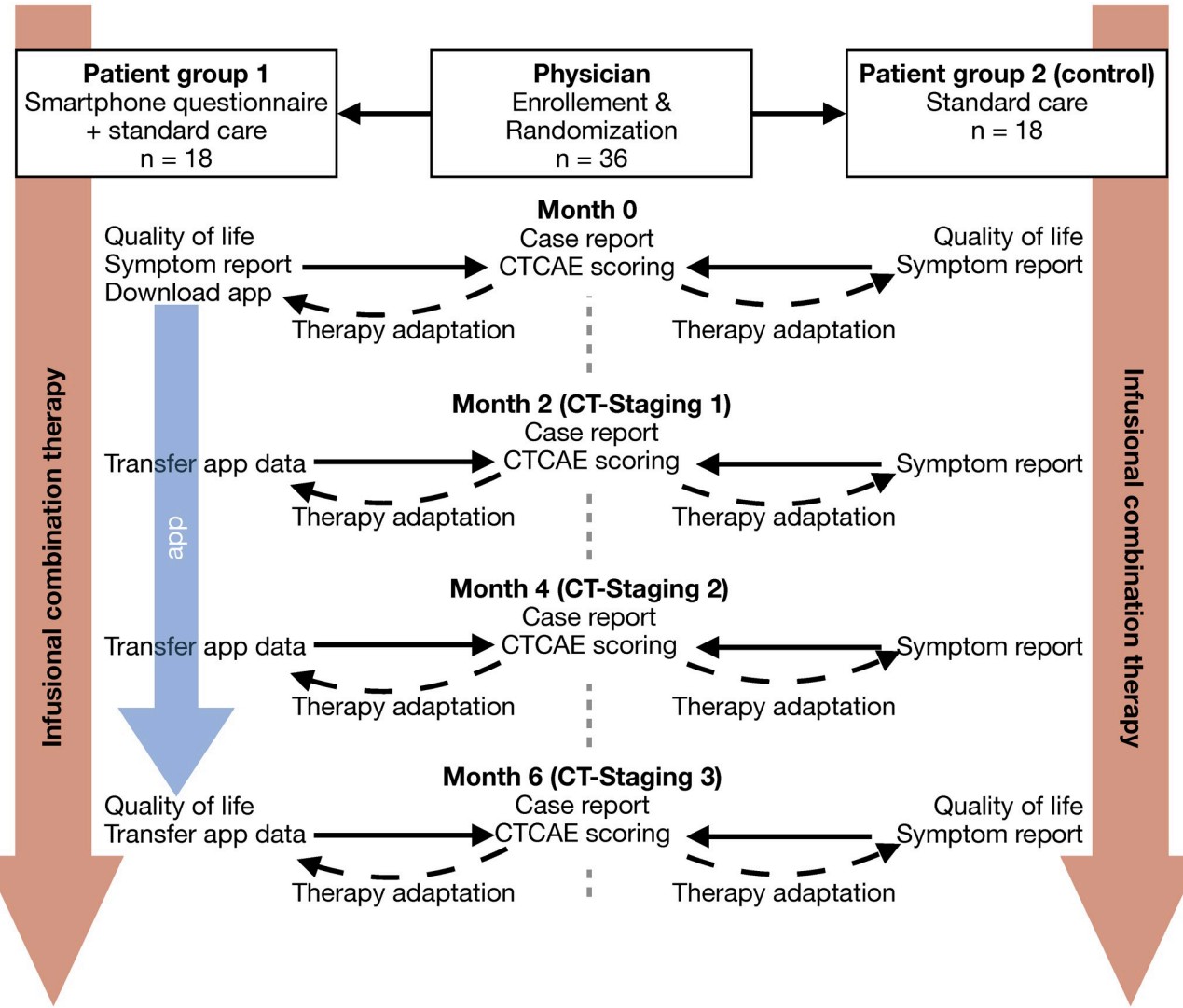

**Fig 1. Second part trial design.** Patients who are treated with infusional combination therapy will report their occurred adverse events (symptoms) every two months to physicians. Group 1 will use a smartphone app with standardized answers to report symptoms while it depends on the patients of group 2 which symptoms they report.

patient remembers and reports depends on each participant and is very individual. Based on these reported adverse events, physicians will delineate a CTCAE scoring. During the reevaluation visit, physicians will additionally fill out an electronic case report form containing tumor stage and information about therapeutic substances and doses. Moreover, patients of both study arms will answer a questionnaire about life quality which is based on the the European Organization for Research and Treatment of Cancer (EORTC) questionnaire [21] at enrolment and once at the end of the study to investigate whether NEMO can improve the quality of life. The design of the second part is shown in Fig 1. The protocol of the study was approved by the Ethikkommission der Universität Ulm (engl.: Ulm University ethical review committee, address: Helmholtzstrasse 20, 89081 Ulm, Germany) on December 11, 2019 (reference 406/19-FSt/bal).

## Participants

Patients over 18 years who receive a tumour therapy of at least two combination partners from chemotherapy, immune therapy and/or targeted therapies at the University Hospital Ulm are invited to participate in the study. Only people with severe neurological or psychiatric disorders which could impair their ability to consent will be excluded from the study. Furthermore, participants have to be able to read, understand, and speak German. A written declaration of informed consent is required to participate in the study.

## Biometry—Sample size estimation

Sample size was determined with the statistical software G*Power (v3.1.9.4) [22]. We assumed an effect size of $d = 1$, corresponding to a side effect variation of 1 on the Likert scale, a type I error rate $\alpha = 0.05$, and a power of 80%. Planning for a Mann-Whitney test comparing the difference between two groups led to a required sample size of 18 individuals per group (Fig 2).

## Questionnaires

The questionnaires are based on literature review and validated instruments including the Common Terminology Criteria for Adverse Events (CTCAE V5.0) and EORTC quality of life questionnaire for cancer patients (QLQ-C30) [21]. Questions for the smartphone application (S1 and S2 Tables) were selected by clinical experts from the list of adverse events of the CTCAE version 5.0. Here, it was focused on frequently appearing side effects that had a strong impact on therapy safety. To adapt the questionnaires more specifically to the respective therapy, a distinction has been made between common adverse events of chemotherapies and immunotherapies and/or targeted therapies. Furthermore, the smartphone questionnaire was extended with questions about wellbeing, the possibility to write a note as well as by a question about the usage of additional medications. In general, possible answers are formulated specifically to keep the scope for interpretation by the participants as small as possible, e.g., how often did you go to the bathroom with diarrhea today? Not at all, 1–3 times, 4–6 times or more than 6 times.

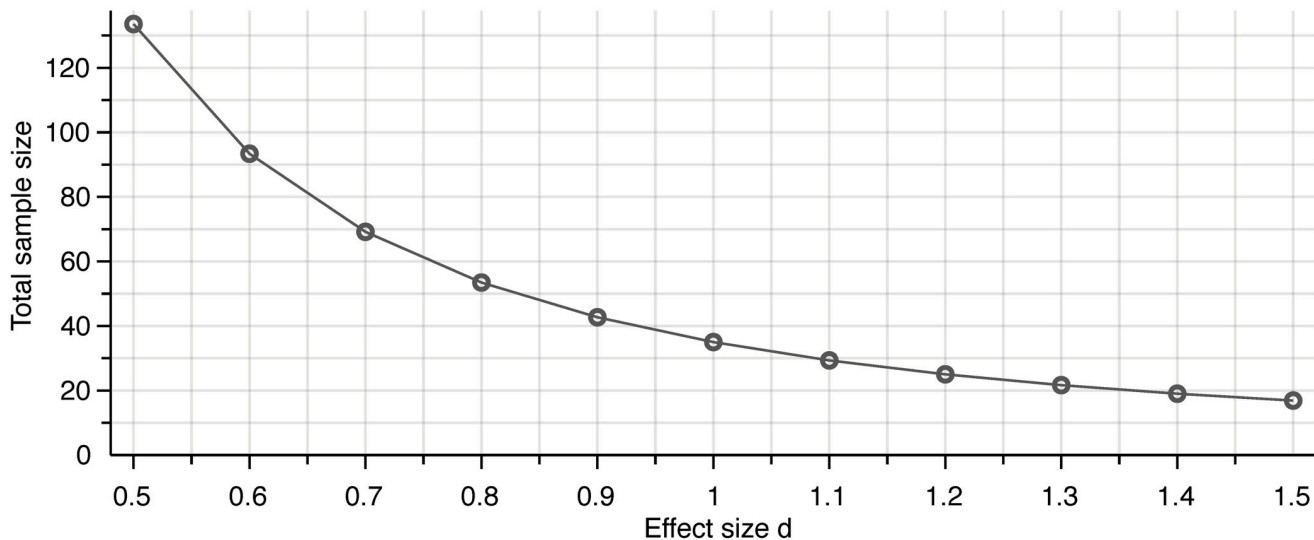

**Fig 2. Sample size estimation.** Sample size calculation for the Mann-Whitney test given an effect size of $d = 1$, a type I error rate $\alpha = 0.05$, a power of 80% led to a required sample size of 36 participants, separated into two groups of 18 individuals each.

The CTCAE scoring ranges from mild symptoms (grade 1) to death (grade 5) [4]. We modified this scoring including no symptom (grade 0) and stopped our scoring at grade 3. This procedure is based on the fact that our questionnaires are designed for outpatients, thus excluding life-threatening consequences (grade 4) or death (grade 5).

To assess the quality of life, the EORTC QLQ-C30 version 3.0 will be used [21].

### Statistical analysis plan

Quality of life questionnaires will be analyzed according to the EORTC-QLQ-C30 manual [23].

Mann-Whitney test will be used to assess known-groups validity, as well as to compare differences between NEMO users and patients with standard reporting procedure concerning the severity of occurred adverse events. NEMO uses a Likert scale to score each questionnaire item. To rate the overall effect of adverse event management, a summary score will be calculated from a weighted sum of questionnaire items by the maximum number of items. Spearman correlation will be used to evaluate convergent validity between each item and the summary score of the quality of life questionnaire or its symptom scales. The impact of the side effect management will be analyzed in the second part by comparing changes in an item's scoring between the reevaluation steps during tumor therapy. For this analysis of responsiveness we plan to test for monotonically decreasing scores within the set of items. This comparison will be made using a Jonckheere-Terpstra test [15, 24].

## Results

Physicians frequently score adverse events during cancer treatment lower than patients do [13, 14]. Nevertheless, these reported symptoms by physicians correlate more to clinical outcomes [15]. We and others believe that discrepancies between reported symptoms may be based on different scoring systems [14]. Therefore, we created an own questionnaire for patient reported symptoms (S1 and S2 Tables) that resemble the clinical scoring routine. These questionnaires were embedded in a smartphone application that can freely be downloaded in all common app stores (ioS, and Android). Likewise the clinical counterpart for visualization is freely available for all current operating systems.

### Oncological applications should be adapted to the needs of elderly

The incidence of developing a tumor disease increases with age and is highest from the age of 70 [25]. For this reason, the smartphone app for the standardized documentation was designed to the needs of the elderly, including clear design, big fonts, and auditory feedback (Fig 3).

Feasibility studies with the elderly showed that many features in mobile applications can irritate them. Therefore, they require menus with shallow navigation depth and avoidance of unseen features [19, 20]. To take these requirements into account we used an always visible menu tab with no further subsections in our app. Since meanings of icons can be misleading [19] we combined them with textual descriptions. Similarly, we designed the questionnaires within the smartphone application (Fig 3). Here, we adjusted the questions and all responding answers to the maximal display size and used high contrasts for the scoring color and font as recommended by Kim et al. [19].

To adjust the performance of older adults, we used buttons with large spacing and big sensitive touch areas. For these issues, it was showed that they can improve reaction times and accuracy of older users [18]. Moreover, we implemented acoustic and visible signals if a button is pressed. On pressing one of the answer options, the user is directly forwarded to the following

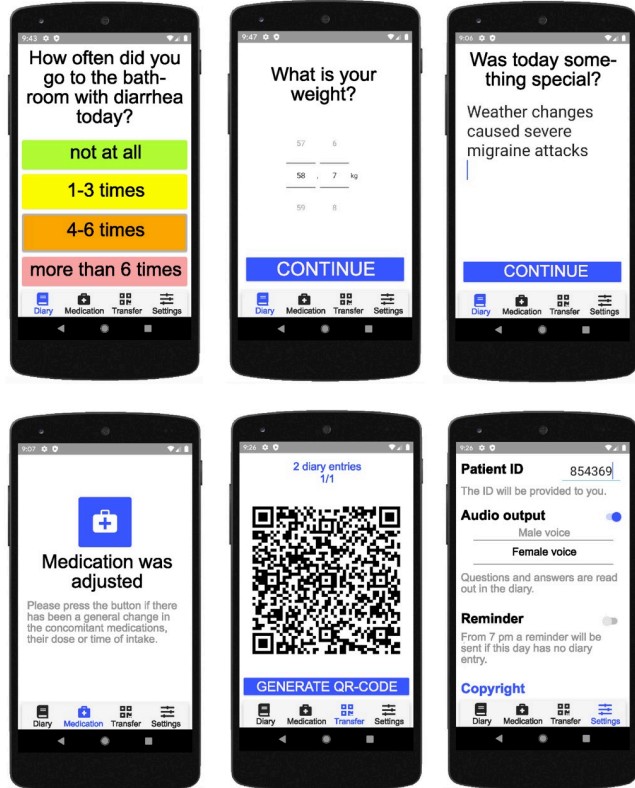

**Fig 3. NEMO smartphone app.** (a) In the settings, users can choose between two therapy schemes and the corresponding questionnaires as well if they want a reminder to daily answer the questionnaire or whether they prefer to hear possible answers with a male or female voice. (b) If the smartphone app opens, the questionnaire about occurred adverse events can be started by clicking on the plus button. (c,d) Each issue is depicted on an independent site and can be passed by answering the question. An obvious color scheme with high contrast indicates the scoring of the side effect. (e) Information about the current weight or pulse can be entered with a predefined picker to avoid misinterpretation by different scales. (f) Furthermore, patients can enter a comment. (g) At the end of the questionnaire, all given answers are summarized and can be changed if the answer is not correct. (h) If the family doctor adjusts the medication, the physician in the clinic can be informed by pressing the button "medication changed". The meaning of buttons and settings is explained by short descriptions (e.g. please press the button if there was a general change for medication).

question. However, errors can be corrected afterward, and a final summary allows the user to check the answers once again before storing them.

Toxic effects of the therapy, e.g. polyneuropathy, accumulate during the therapy [14]. In the event of treatment-associated deterioration of the general condition, it is possible to be guided by acoustic navigation. This offers support for the patients to continue answering the questions even if they feel worse.

## Secure data transfer and patient empowerment

Data protection had a top priority for the development of NEMO. For this purpose, the smartphone app runs exclusively offline and consequently avoids transmission over potentially insecure internet connections to store data on remote servers. Likewise, transmission via Bluetooth or mails includes security risks [26]. Consequently, patient-reported symptoms are soley stored locally and can only be transmitted on sight with the physician in charge. We developed a protocol based on quick response (QR) codes that allow transmission only by direct visual

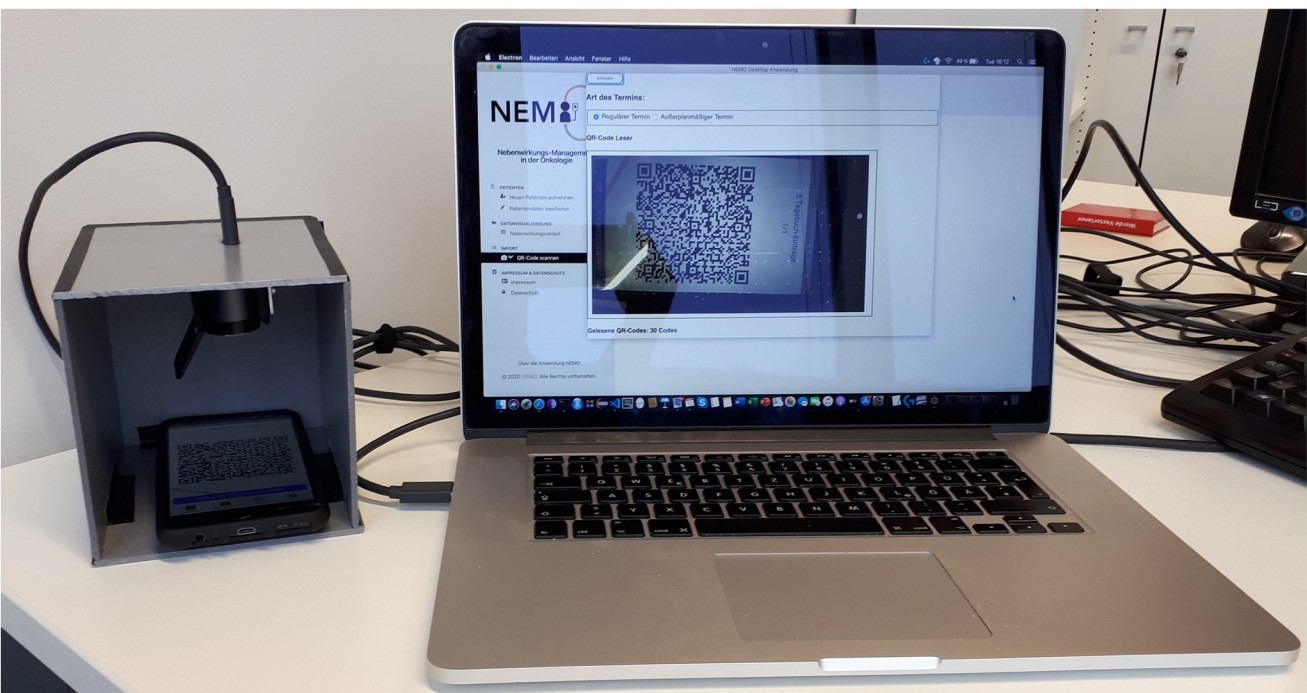

**Fig 4. Secure data transfer.** If a patient is willing to transfer its data, a QR code can be generated on the mobile phone (a) that allows secure data transfer via scanning (b). Example of the data transfer from the mobile phone to the desktop computer (image copyright Institute of Medical Systems Biology, Ulm University).

contact (Fig 4). This transmission excludes security risks associated with other devices in closer proximity [26]. The patient retains control over the data, which increases overall compliance and, in a broader sense, therapy safety.

The data protection concept was positively reviewed by the data protection officer of the University Hospital Ulm and the federal-state data protection officer and is in accordance with the General Data Protection Regulation (GDPR).

### Symptom visualization for physicians

Physicians often have limited time resources for each patient and this may influence the assessment of occurring therapy side effects. Fast and easy visualization of therapy-related adverse events are very important for monitoring and adjusting the oncology therapy. The NEMO desktop app provides a detailed visualization of all occurred adverse events per day as well as over time (Fig 5). The events can be presented day-by-day or over a longer period, allowing a more individual adaptation of the therapy as well as an optimized supportive therapy. Furthermore, physicians can choose between a line plot (Fig 5a) or a heat map visualization (Fig 5b). This allows patient individualized adverse event tracking over time. Medical appointments or therapy changes are highlighted on the time axis. On hovering over specific eventy, detailed information pops up.

Besides the management of patients involved in this study, the NEMO desktop application allows the documentation of medication and therapy doses. The application also considers the possibility to export PROs into other clinic information systems to support clinical documentation.

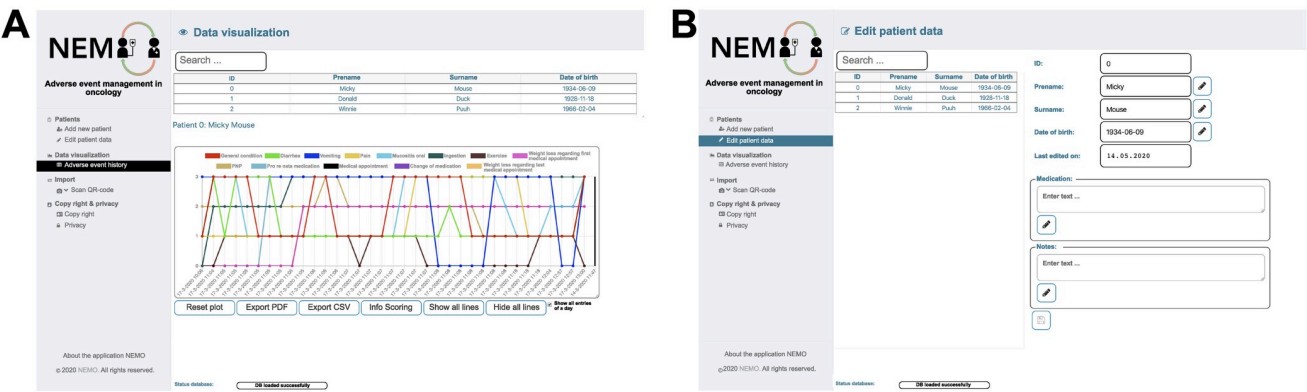

**Fig 5. Visualization of transferred adverse events.** The physician in charge can display patients' transferred data of occurred adverse events as a line plot (a) or heat map (b). This allows a fast and easy analysis.

## Discussion

Feasibility studies with various PROs are well described in the literature. Most of these studies investigated whether patients cope with questionnaires on adverse events [6–8, 27] and how regularly these questionnaires were filled out [6, 9, 27]. Others further included evaluations for missing values [2, 13, 27] or studies on the format preferred by patients [2, 8]. Additionally, studies are performed to assess the discrepancy between patients' and clinician's reported symptoms. Here, it was observed that physicians score adverse events during cancer treatment frequently lower than patients do [13, 14]. However, to the best of our knowledge, none of the performed studies showed results whether considering and treating patient-reported adverse events will enable clinicians to adjust the therapy individually. This prompt treatment adjustment would help to achieve the highest possible effectiveness together with therapeutic safety. Toxic effects due to therapy are often cumulative and became more and more prominent in the course of treatment [14]. As follows, it is often necessary to reduce the dosage of treatment that can be applied in the course of therapy. This could be avoided if these adverse events could be treated by early interventions. In this study, we aim to use a digital approach and CTCAE-adapted questions for patient-reported symptoms to improve the therapeutic process and to pave the way towards personalized medicine.

Another important issue for patient compliance is their empowerment. Reported treatment symptoms are private data and thus require high protection level. Therefore, a secure data transfer was also the focus of the implementation. This excludes data transfer via mails or potential insecure connections in order to store data on remote servers. Instead, the graduation of symptoms are stored locally on the patient's smartphone. The app does not store any further personalized data, such as name. Patients can then decide to send their data on sight via a QR code-based communication protocol to the physician in charge. Consequently, the patients are always in control of their data and their sharing. Transferred data is saved on a secure and backuped server in the clinic.

In contrast to previously performed PRO studies that have a recall time of seven days [2, 6], we request the participants to daily answering the questionnaire of the smartphone application. Thereby, we want to map also minor changes of symptoms that are usually not noticed and which can lead to early adjustments of the therapy. Moreover, it might be easier for patients to describe their current adverse events instead of recalling and evaluating symptoms that change significantly over the course of the week. We are aware that daily answering is

time consuming. Therefore, we limited ourselves to the most common adverse events and questions about well being which are essential for therapy success. Thus, in total we ask twelve or 21 questions depending on the underlying therapy scheme. Since patients already have to report occurred adverse events to their physicians, we believe that standarized guided answers will support them. Additionally, detailed and standardized descriptions of PRO will improve the comparability between views of patients and clinician. In order to support patients in their adherence to therapy, it is possible for patients to receive a reminder from the NEMO application that they have not yet created an entry on the respective day. Thereby, we hope to reduce the number of missing data points. However, we are aware of the possibility that a daily preoccupation with one's illness could lead to an over-perception of the symptoms. This issue will be addressed in the second part of our study.

## Conclusion

This is one of the first approaches that aims at an in-depth feasibility analysis of PROs in the clinical routine. By using smartphone-based questionnaires with standardized requests for common adverse events in oncological treatments we aim to improve therapy safety due to a faster management of side effects. This way of tracking of adverse events may allow a more personalized care and a possible better tolerability, reflecting a better general condition for patients which will be investigated in the clinical trial. Mainly, we plan to investigate the potential of a better adverse event management due to smartphone application in support of cancer therapies. However, collected data can later be applied to generate event prediction models. Besides, this approach may improve intersectoral communication and patient care in the long term. Family doctors are often the first point of contact for patients if adverse events occur during tumor therapy. However, they often do not have sufficient access to the relevant data, so the app can act as a bridge of information transfer and help in the current treatment situation.

## Supporting information

**S1 File.**
(DOCX)

**S2 File.**
(DOCX)

**S3 File.**
(DOCX)

**S1 Table. Smartphone questionnaire for chemotherapies.** The grades refer to the severity of the occurred adverse effect. The CTCAE scoring includes grade 1 to 5. We modify this scoring including grade 0 (= no side effect) and stopped our scoring at grade 3. This procedure is based on the fact that our app is designed for outpatients, thus excluding life-threatening consequences (grade 4) or death (grade 5). The meaning of each grading is in accordance with (CTCAE v5.0 Clean, Tracked, and Mapping Document) and was translated into German. (PDF)

**S2 Table. Smartphone questionnaire for immunotherapies.** The grades refer to the severity of the occurred adverse effect. The CTCAE scoring includes grade 1 to 5. We modify this scoring including grade 0 (= no side effect) and stopped our scoring at grade 3. This procedure is based on the fact that our app is designed for outpatients, thus excluding life-threatening consequences (grade 4) or death (grade 5). The meaning of each grading is in accordance with

(CTCAE v5.0 Clean, Tracked, and Mapping Document)and was translated into German.
(PDF)

**S3 Table. Questions for smartphone feasibility.**
(PDF)

## Author Contributions

**Conceptualization:** Angelika M. R. Kestler, Hans A. Kestler.

**Funding acquisition:** Hans A. Kestler.

**Methodology:** Angelika M. R. Kestler, Silke D. Kühlwein, Johann M. Kraus.

**Project administration:** Hans A. Kestler.

**Resources:** Hans A. Kestler.

**Software:** Silke D. Kühlwein, Julian D. Schwab, Robin Szekely, Patrick Thiam, Rolf Hühne, Niels Jahn, Axel Fürstberger, Rainer Schuler.

**Supervision:** Thomas Seufferlein, Hans A. Kestler.

**Visualization:** Silke D. Kühlwein, Julian D. Schwab, Robin Szekely, Patrick Thiam.

**Writing – original draft:** Angelika M. R. Kestler, Silke D. Kühlwein, Johann M. Kraus, Thomas Seufferlein, Hans A. Kestler.

**Writing – review & editing:** Julian D. Schwab, Axel Fürstberger, Nensi Ikonomi, Julien Balig, Peter Kuhn, Florian Steger.

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
