## [Decision Letter · Decision Letter 0]

15 Dec 2020

PONE-D-20-29066

Digitalization of adverse event management in oncology to improve treatment outcome

PLOS ONE

Dear Dr. Kestler,

Thank you for submitting your manuscript to PLOS ONE. After careful consideration, we feel that it has merit but does not fully meet PLOS ONE’s publication criteria as it currently stands. Therefore, we invite you to submit a revised version of the manuscript that addresses the points raised during the review process.

Three reviewers (two experts in the field and one statistical reviewer) assessed your manuscript and have several minor comments. Please ensure that you address all of these in your revision.

We look forward to receiving your revised manuscript.

Kind regards,

Susan Hepp

Academic Editor

PLOS ONE

Journal Requirements:

2. Thank you for stating in your ethics statement on your online submission form

"Ethics committee Ulm University, Ulm, Germany; Proposal no. 406/19: "This concludes the evaluation by the research ethics committee of the University of Ulm with a positive opinion. Consent was obtained in written form." For studies reporting research involving human participants, PLOS ONE requires authors to confirm that this specific study was reviewed and approved by an institutional review board (ethics committee) before

the study began. Please clarify whether this study protocol was specifically approved by your ethics committee.

3. The text of the manuscript states that the protocol for phase II was approved by your ethics committee. Please clarify if the protocol for phase I was also approved by the committee.

4. Please provide additional details regarding participant consent. In the ethics statement in the Methods and online submission information, please ensure that you have specified whether consent was informed.

5. To meet our data availability standards, please provide a link  for the smartphone/desktop application described in this study, or provide detailed instructions as to how another researcher could access it.

6.Please amend either the abstract on the online submission form (via Edit Submission) or the abstract in the manuscript so that they are identical.

Reviewers' comments:

Reviewer's Responses to Questions

**Comments to the Author**

1. Does the manuscript provide a valid rationale for the proposed study, with clearly identified and justified research questions?

Reviewer #1: Yes

Reviewer #2: Yes

Reviewer #3: Yes

2. Is the protocol technically sound and planned in a manner that will lead to a meaningful outcome and allow testing the stated hypotheses?

Reviewer #1: Yes

Reviewer #2: Yes

Reviewer #3: Yes

3. Is the methodology feasible and described in sufficient detail to allow the work to be replicable?

Reviewer #1: Yes

Reviewer #2: No

Reviewer #3: Yes

4. Have the authors described where all data underlying the findings will be made available when the study is complete?

Reviewer #1: Yes

Reviewer #2: Yes

Reviewer #3: No

5. Is the manuscript presented in an intelligible fashion and written in standard English?

Reviewer #1: Yes

Reviewer #2: Yes

Reviewer #3: No

6. Review Comments to the Author

You may also provide optional suggestions and comments to authors that they might find helpful in planning their study.

Reviewer #1: The authors present the protocol of a study obtaining Patient-Reported-Outcomes (PRO) and especially CTCAE criteria. The manuscript is thouroughly written (please note that I am not a native speaker myself). The different sections of the manuscript do not leave much room for improvement. The proposed study is feasible, methods are adeqante and most important the research question and the study is meaningful and will provide patient relevant data.

I only have a few comments. I checked for accuracy conconrinng to the equator-network guidelines. Here, the title should be in line with the recommendations. I would suggest adding a short sentence as second part ouf the titel as "A protocol for a non-randomized prospective feasibility study.

In the discussion and part of the introduction the authors are not precide what really is their underlying aim. For me, it is specifically in screening and early detection of CTCAE criteria. This can be stressed as one of the unique selling points of this study. You may refer as an example to another feasibility study (1) to underline that other studies have merely looked at PROs (a very vast range), but your unique selling point is, again, the adherance to CTCAE criteria.

Please note, that I have not checked manuscript- and references according to the journal style, but leave this to the editorial office.

1 Benze G et al. PROutine: a feasibility study assessing surveillance of electronic patient reported outcomes and adherence via smartphone app in advanced cancer. Ann Palliat Med. 2019 Apr;8(2):104-111. )

Reviewer #2: Important note: This review pertains only to ‘statistical aspects’ of the study and so ‘clinical aspects’ [like medical importance, relevance of the study, ‘clinical significance and implication(s)’ of the whole study, etc.] are to be evaluated [should be assessed] separately/independently. Further please note that any ‘statistical review’ is generally done under the assumption that (such) study specific methodological [as well as execution] issues are perfectly taken care of by the investigator(s). This review is not an exception to that and so does not cover clinical aspects {however, seldom comments are made only if those issues are intimately / scientifically related & intermingle with ‘statistical aspects’ of the study}. Agreed that ‘statistical methods’ are used as just tools here, however, they are vital part of methodology [and so should be given due importance].

COMMENTS: I highly appreciate the study [particularly the objective [Early detection and treatment of adverse events in oncological treatment to improve patients’ safety and outcomes]. However, I have few small {really very ‘minor’} queries. Those are given below. Hope, you will find them worth considering.

1. Although your ‘Article Type: Registered Report Protocol’ it is not indicated anywhere that it is a ‘Protocol’. Preferably the word ‘Protocol’ should appear in title itself. Mention at least in ‘abstract’ is expected.

2. Since ‘Phase I’ is only to assess the feasibility of the application why patients divided into three age groups (<55years, 55-75years, >75years). Do you wish to assess the feasibility of the application for different age groups? You also say that ‘equal-sized’ groups. How was that insured? Use of ‘quota’ sampling may be mentioned.

3. According to ‘abstract-methods’ phase II, will compare the impact of smartphone reported adverse events regarding applied therapy doses and quality of life to those of patients receiving standard care. Here, 36 patients will be randomly assigned to each of the two groups. Which two groups is not described there? It is clear only from figure-2 that ‘Patients in group 1: Smartphone questionnaire + standard care’ and Patients in group 2 (control) : Standard care’.

4. In figure-2 Y-axis heading is ‘Total Sample Size’ but generally sample size is given ‘per arm’. I request you to kindly confirm {May please refer to line 124}. Figure-2 is 100% correct, but modified/adjusted/derived.

5. Supporting information files are not named as ‘S1 Table. Smartphone questionnaire for chemotherapies’, ‘S2 Table. Smartphone questionnaire for immunotherapies’ and ‘S3 Table. Feasibility analysis’ as indicated in manuscript. Therefore, ‘S3 Table. Feasibility analysis’ is not available [which is important for me].

6. Refer to section ‘Biometry - sample size estimation’ as well as to lines 148-49 which says regarding Wilcoxon-Mann-Whitney test [Wilcoxon-Mann-Whitney test will be used to assess known-groups validity, as well as 148 to compare differences between NEMO users and patients with standard reporting procedure concerning the severity of occurred adverse events] however, note that ‘Wilcoxon’s Rank Sum’ test and ‘Mann-Whitney U’ test are separate tests [though equivalent]. There is no test by all three together {test called ‘Wilcoxon-Mann-Whitney test’} in my knowledge. Kindly check. (scientist Wilcoxon had given one more test called ‘Wilcoxon’s Signed Ranked test’ which is for paired samples)

7. If feasibility study is already completed [‘S3 Table. Feasibility analysis’ indicates so] then why ‘Phase I’ [which you say is to assess the feasibility of the application] because anyway in this article few sections are only for phase II [though not specified so (which is not desirable, I guess) ex.: section ‘Biometry - sample size estimation’]. Inclusion of ‘Phase I’ may increase confusion.

8. Further comment on (more account but very briefly) on ‘Jonckheere-Terpstra test’ is required, I guess (line 157). Why do you need ‘A distribution-free k-sample test against ordered alternatives’?

Reviewer #3: This an interesting area of research.

1. It is not clear from the title that this paper describes the design of a future trial using the new data collection app that the authors have designed. The title and abstract should be changed to make this more obvious.

2. As noted above, the paper describes a future trial. However, it also describes a tool for collecting adverse events using a smartphone app. This app is novel but its development is not described in enough detail. How does AE collection using the app compare with standard AE data collection? Do patients and clinicians like the app? How exactly did they participate in the app development?

3. The figures showing the app and the "clinician dashboard" are in German. Since most readers of the medical literature may not understand German, the authors need to consider an english language version of the figures.

7. PLOS authors have the option to publish the peer review history of their article (what does this mean?). If published, this will include your full peer review and any attached files.

Reviewer #1: No

Reviewer #2: **Yes: **Dr. Sanjeev Sarmukaddam

Reviewer #3: No

---

## [Author Response · Author response to Decision Letter 0]

24 Dec 2020

Please find our point-to-point response attached.

---

## [Decision Letter · Decision Letter 1]

17 May 2021

Digitalization of adverse event management in oncology to improve treatment outcome - A prospective study protocol

PONE-D-20-29066R1

Dear Dr. Kestler,

We’re pleased to inform you that your manuscript has been judged scientifically suitable for publication and will be formally accepted for publication once it meets all outstanding technical requirements.

Kind regards,

Jeremy Yuen Chun Teoh

Academic Editor

PLOS ONE

Reviewers' comments:

Reviewer's Responses to Questions

**Comments to the Author**

1. Does the manuscript provide a valid rationale for the proposed study, with clearly identified and justified research questions?

Reviewer #1: Yes

Reviewer #2: Yes

Reviewer #3: Yes

2. Is the protocol technically sound and planned in a manner that will lead to a meaningful outcome and allow testing the stated hypotheses?

Reviewer #1: Yes

Reviewer #2: Yes

Reviewer #3: Yes

3. Is the methodology feasible and described in sufficient detail to allow the work to be replicable?

Reviewer #1: Yes

Reviewer #2: Yes

Reviewer #3: Yes

4. Have the authors described where all data underlying the findings will be made available when the study is complete?

Reviewer #1: Yes

Reviewer #2: Yes

Reviewer #3: Yes

5. Is the manuscript presented in an intelligible fashion and written in standard English?

Reviewer #1: Yes

Reviewer #2: Yes

Reviewer #3: Yes

6. Review Comments to the Author

You may also provide optional suggestions and comments to authors that they might find helpful in planning their study.

Reviewer #1: Thank you for provision of the thouroughly prepared point-by-point table. The changes are sound and meaningful.

Reviewer #2: COMMENTS: Since all the comments made on earlier draft by me (and hopefully by other respected reviewers also) are attended positively/adequately, I am fully satisfied and the manuscript is improved a lot.

Only thing noticed now is about the ‘abstract’. Actually, your ABSTRACT is very well drafted but assay type. Please note that it is preferable [refer to item 1b of CONSORT checklist 2010: Structured summary of trial design, methods, results, and conclusions] to divide the ABSTRACT with small sections like ‘Objective(s)’, ‘Methods’, ‘Results’, ‘Conclusions’, etc. which is an accepted practice of most of the good/standard journals [including this one]. It will definitely be more informative then, I guess, whatever the article type may be. As this article is on ‘protocol’, section headings may differ which does not matter.

I recommend acceptance (with this minor revision).

Reviewer #3: Review Comments to the Author

You may also provide optional suggestions and comments to authors that they might find helpful in planning their study.

(Please upload your review as an attachment if it exceeds 20,000 characters) (Limit 100 to 20000 Characters)

"YOU HAVE ANSWERED MY QUESTIONS AND THE CURRENT MANUSCRIPT IS USEFUL AND WELL WRITTEN."

7. PLOS authors have the option to publish the peer review history of their article (what does this mean?). If published, this will include your full peer review and any attached files.

Reviewer #1: No

Reviewer #2: **Yes: **Dr. Sanjeev Sarmukaddam

Reviewer #3: No

---

## [Editor Report · Acceptance letter]

27 May 2021

PONE-D-20-29066R1 

Digitalization of adverse event management in oncology to improve treatment outcome - A prospective study protocol 

Dear Dr. Kestler:

I'm pleased to inform you that your manuscript has been deemed suitable for publication in PLOS ONE. Congratulations! Your manuscript is now with our production department. 

Kind regards, 

on behalf of

Dr. Jeremy Yuen Chun Teoh 

Academic Editor

PLOS ONE